# Clustering huge protein sequence sets in linear time

Martin Steinegger [1,2,3] & Johannes Söding [1]

Metagenomic datasets contain billions of protein sequences that could greatly enhance large-scale functional annotation and structure prediction. Utilizing this enormous resource would require reducing its redundancy by similarity clustering. However, clustering hundreds of millions of sequences is impractical using current algorithms because their runtimes scale as the input set size $N$ times the number of clusters $K$, which is typically of similar order as $N$, resulting in runtimes that increase almost quadratically with $N$. We developed Linclust, the first clustering algorithm whose runtime scales as $N$, independent of $K$. It can also cluster datasets several times larger than the available main memory. We cluster 1.6 billion metagenomic sequence fragments in 10 h on a single server to 50% sequence identity, >1000 times faster than has been possible before. Linclust will help to unlock the great wealth contained in metagenomic and genomic sequence databases.

---

[1] Quantitative and Computational Biology group, Max-Planck Institute for Biophysical Chemistry, Am Fassberg 11, 37077 Göttingen, Germany. [2] Department for Bioinformatics and Computational Biology, Technische Universität München, 85748 Garching, Germany. [3] Department of Chemistry, Seoul National University, 08826 Seoul, Republic of Korea. Correspondence and requests for materials should be addressed to M.S. (email: martin.steinegger@mpibpc.mpg.de) or to J.S. (email: soeding@mpibpc.mpg.de)

In metagenomics, DNA is sequenced directly from the environment, allowing us to study the vast majority of microbes that cannot be cultivated in vitro[1]. During the last decade, costs and throughput of next-generation sequencing have dropped two-fold each year, twice faster than computational costs. This enormous progress has resulted in hundreds of thousands of metagenomes and tens of billions of putative gene and protein sequences[2,3]. Therefore, computing and storage costs are now dominating metagenomics[4–6]. Clustering protein sequences predicted from sequencing reads or pre-assembled contigs can considerably reduce the redundancy of sequence sets and costs of downstream analysis and storage.

CD-HIT and UCLUST[7,8] are by far the most widely used tools for clustering and redundancy filtering of protein sequence sets (see ref. [9] for a review). Their goal is to find a representative set of sequences such that each of the input set sequences is represented well enough by one of the $K$ representatives, where "well enough" is quantified by some similarity criteria.

Like most other fast sequence clustering tools, they use a fast prefilter to reduce the number of slow pairwise sequence alignments. An alignment is only computed if two sequences share a minimum number of identical $k$-mers (substrings of length $k$). If we denote the average probability by $p_{match}$ that this happens by chance between two non-homologous input sequences, then the prefilter would speed up the sequence comparison by a factor of up to $1/p_{match}$ at the expense of some loss in sensitivity. This is usually unproblematic: if sequence matches are missed (false negatives) we create too many clusters, but we do not lose information. In contrast, false positives are costly as they can cause the loss of unique sequences from the representative set.

CD-HIT and UCLUST employ the following "greedy incremental clustering" approach: each of the $N$ input sequences is compared with the representative sequences of already established clusters. When the sequence is similar enough to the representative sequence of one of the clusters, that is, the similarity criteria such as sequence identity are satisfied, the sequence is added to that cluster. Otherwise, the sequence becomes representative of a new cluster. Due to the comparison of all sequences with the cluster representatives, the runtimes of CD-HIT and UCLUST scale as O($NK$), where $K$ is the final number of clusters. In protein sequence clustering $K$ is typically of similar size to $N$ and therefore the total runtime scales almost quadratically with $N$. The fast sequence prefilters speed up each pairwise comparison by a large factor $1/p_{match}$ but cannot improve the time complexity of O($NK$). This almost quadratic scaling results in impractical runtimes for a billion or more sequences.

Here we present the sequence clustering algorithm Linclust, whose runtime scales as O($N$), independent of the number of clusters found. We demonstrate that it produces clusterings of comparable quality as other tools that are orders of magnitude slower and that it can cluster over a billion sequences within hours on a single server.

## Results

**Overview of Linclust.** The Linclust algorithm is explained in Fig. 1 (for details see Methods and Fig. 5). As in previous methods, we reduce the number of pairwise comparisons by requiring the sequences to share at least one identical $k$-mer substring. A critical insight to achieve linear time complexity is that we need not align every sequence with every other sequence sharing a $k$-mer (see steps 3,4). We reach similar sensitivities by selecting only a very small subset of sequences as "center sequences" (colored dots) and only aligning sequences to the center sequences with which they share a $k$-mer. Linclust thus

requires less than $mN$ sequence comparisons with a small constant $m$ (default value 20), instead of the $\sim NKp_{match}$ comparisons needed by UCLUST, CD-HIT and other tools.

In most clustering tools, the main memory size severely limits the size of the datasets that can be clustered. UCLUST, for

(1) Select $m$ (e.g. 20) $k$-mers per sequence and find groups of sequences sharing a $k$-mer.

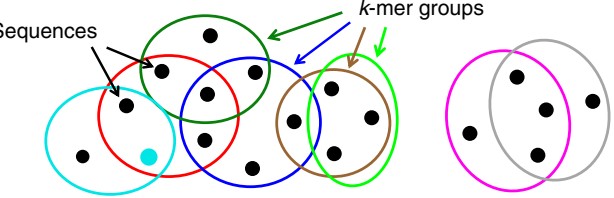

(2) Select longest sequence per group as center sequence

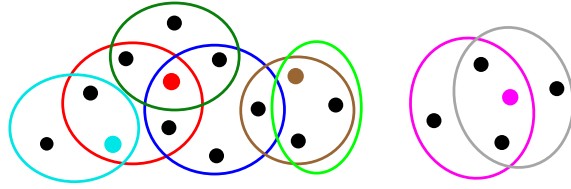

(3,4) Compare each sequence in group only with center sequence…

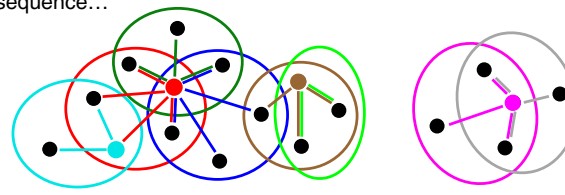

… not with all sequences in the group

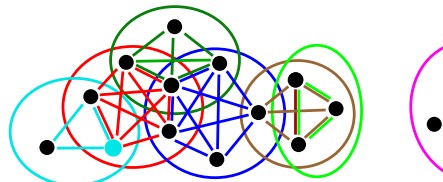

(5) Sequences are recruited by center sequences into clusters

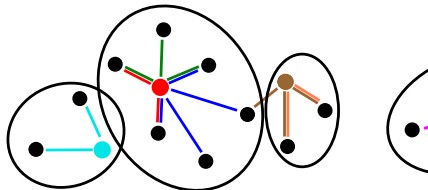

**Fig. 1** Overview of linear-time clustering algorithm. (1) For each sequence Linclust selects $m$ $k$-mers (with the lowest hash function values). It sorts the $k$-mers alphabetically in quasi-linear time to find the groups of sequences sharing a $k$-mer (colored sets) and (2) it selects the longest sequence per $k$-mer group as center. (3,4) It compares each sequence (in three consecutively slower and more sensitive steps) only with the center sequences it shares a $k$-mer with, not with all sequences it shares a $k$-mer with. It therefore needs to compute at most $m$ comparisons per sequence and $mN$ in total. Pairs that pass the clustering criteria are linked by an edge. (5) The sequences are clustered in time O($mN$) using a greedy incremental algorithm that finds clusters whose members all have an edge with a representative sequence. For a more details see Fig. 5

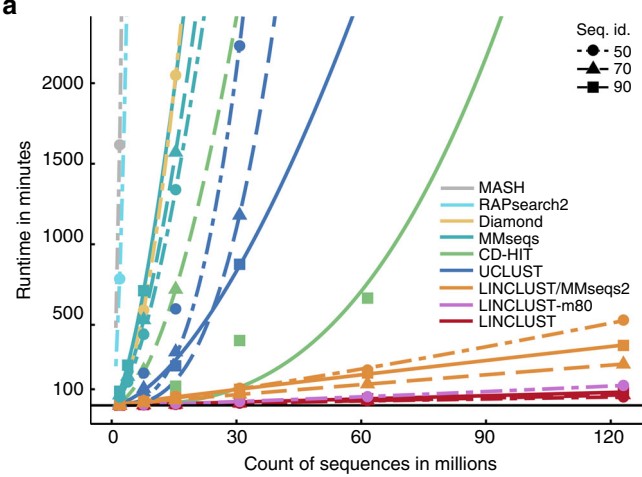

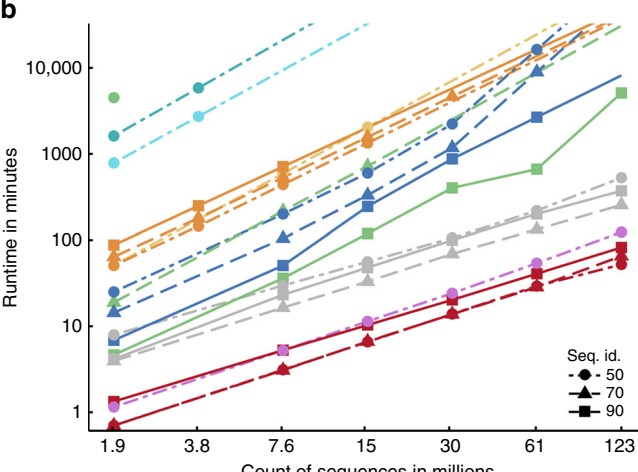

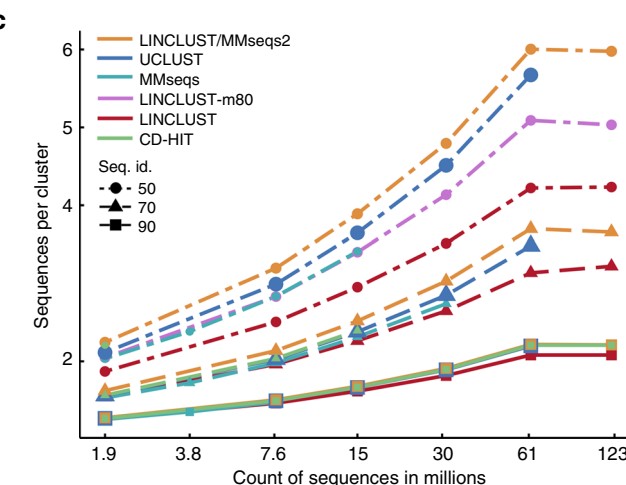

**Fig. 2** Linclust and Linclust/MMseqs2 manifest unique linear scaling of runtime with sequence set size. **a** Runtime versus input set size on linear scales. The plotting symbols indicate the sequence identity threshold for clustering of 90%, 70%, and 50%. The curves are fits with a power law, $bN^a$. For comparison, we include runtimes of all-against-all searches using sequence search tools DIAMOND, RAPsearch2, and MASH. Runtimes were measured on a server with two Intel Xeon E5-2640v3 8-core CPUs and 128 GB RAM. **b** Same as (**a**) but on log-log scales. **c** Average number of sequences per cluster at 90%, 70%, and 50% sequence identity. Larger average cluster sizes imply higher sensitivities to detect similar sequences

example, needs 10 bytes per residue of the representative sequences. Linclust needs $m \times 16$ bytes per sequence, but before running it automatically checks available main memory and if necessary splits the table of $mN$ lines into chunks such that each chunk fits into memory (Supplementary Fig. 1 and Methods). It then processes the chunks sequentially. In this way, Linclust can cluster sequence sets that would occupy many times its main memory size at almost no loss in speed.

**Linclust and Linclust/MMseqs2 workflows**. We integrated Linclust into our MMseqs2 (Many-versus-Many sequence searching) software package[10], and we test two versions of Linclust in our benchmark comparison: the bare Linclust algorithm described in Fig. 1 (simply named "Linclust"), and a combined four-step cascaded clustering workflow ("Linclust/MMseqs2"). In this workflow, a Linclust clustering step is followed by one (above 60% sequence identity) or three (≤60%) clustering steps, each of which clusters the representative sequences from the previous step by performing an increasingly sensitive all-against-all MMseqs2 sequence search followed by the greedy incremental clustering algorithm. We also include in our benchmark our original MMseqs clustering tool[11].

**Runtime and clustering sensitivity benchmark**. We measure clustering runtimes on seven sets: the 61522444 sequences of the UniProt database, randomly sampled subsets with relative sizes 1/16, 1/8, 1/4, 1/2, and UniProt plus all reversed sequences (123 million sequences). Each tool clustered these sets using a minimum pairwise sequence identity of 90%, 70% and 50%. Sequence identity was defined similarly for all three tools. The three tools use somewhat different strategies to try to ensure that only proteins with the same domain architecture are clustered together (see Methods: Clustering criteria).

At 50% identity, Linclust clusters the 123 million sequences 10 times faster than Linclust/MMseqs2 and, by extrapolation, 2300 times faster than UCLUST, 720 times faster than MMseqs, 4600 times faster than CD-HIT, 1600 times faster than DIAMOND[12], 69000 times faster than MASH[13], and 26000 times faster than RAPsearch2[14] (Fig. 2a, b). At 90% identity, Linclust still clusters these sequences 570 times faster than MMseqs, 100 times faster than UCLUST, 62 times faster than CD-HIT, and 4.5 times faster than Linclust/MMseqs2.

At 90% sequence identity threshold, we determined how the runtimes scale with the input set size $N$ by fitting a power law $\left(T \sim aN^b\right)$ to the measured runtimes. Runtimes scale very roughly quadratically for UCLUST ($N^{1.62}$) and CD-HIT ($N^{2.75}$) whereas they grow only linearly for Linclust/MMseqs2 ($N^{0.94}$) and Linclust ($N^{1.01}$). The speed-ups due to Linclust's Hamming distance stage and the ungapped alignment filter are analyzed in Supplementary Fig. 2.

To assess the clustering sensitivity, we compare the average size of clusters: a deeper clustering with more sequences per cluster implies a higher sensitivity to detect similar sequences. All three tools produce similar numbers of clusters at 90% and 70% sequence identity (Fig. 2c). Importantly, despite Linclust's linear scaling of the runtime with input set size, it manifests no loss of sensitivity for growing dataset sizes. At 50%, Linclust produces 13% more clusters than UCLUST. But we can increase Linclust's sensitivity simply by selecting more $k$-mers per sequence. By increasing $m$ from 20 to 80, Linclust takes only 1.5 to 2 times longer but attains a sensitivity similar to UCLUST (pink in Fig. 2a–c, Supplementary Fig. 4).

To estimate the fraction of missed sequence pairs that could have been clustered together, we examined the distribution of sequence identities between representative cluster sequences

(Fig. 3a–c). For each clustering run, we searched with BLAST[15] a random sample of 1000 representative sequences against all representative sequences of the clustering. We show the cumulative distribution of sequence identities for the best matches that satisfy the minimum coverage threshold of 90% used in the clustering runs. This coverage threshold is favorable for UCLUST since its own coverage criterion is less strict (see Methods, "Clustering criteria"). Due to the heuristic prefiltering methods employed by all tools, none produces a perfect clustering. This limitation is seen most clearly at 50% sequence identity (Fig. 3c), for which Linclust/MMseqs2, UCLUST, Linclust-m80 and Linclust miss 2%, 10%, 16% and 28% of sequence pairs satisfying the clustering threshold.

**Cluster consistency analysis**. We measure the quality of the clusterings produced by the tools by analyzing the homogeneity of the functional annotation of the sequences in the clusters[16]. We assess Gene Ontology (GO) annotations[17] (Fig. 4a, b) and Pfam domain annotations[18] (Fig. 4c) provided by the UniProt database. For each of these annotations, we averaged two score variants over all clusters, "mean" and a "worst". The "mean" ("worst") score for a cluster is the mean (minimum) annotation similarity score between the representative sequence and all other cluster members, as described in ref. [16].

Overall, the consistencies of cluster annotations are similar for all tools, which is not surprising since they all use exact Smith-Waterman alignments and similar acceptance criteria (Supplementary Fig. 3, Methods). However, Linclust/MMseqs2 and Linclust clusterings have better consistencies than UCLUST and CD-HIT according to purely experimentally derived GO annotations (Fig. 4a) and according to Pfam domain annotations (Fig. 4c). This might be either due to a stricter minimum coverage criterion in Linclust or due to its slightly different definition of sequence similarity, which translates the sequence identity threshold into an approximately equivalent threshold for the similarity score of the local alignment divided by the maximum length of the two aligned segments (Methods: Clustering criteria). This similarity measure is more appropriate than sequence identity to cluster together sequences with conserved functions, as it also accounts for gaps and for the degree of similarity between aligned residues. The cluster consistencies of all tools are similar when GO annotations based on computational predictions are included (Fig. 4b).

**Clustering 1.6 billion metagenomic sequences**. As a demonstration of Linclust's ability to cluster huge sets, we applied it to cluster 1.59 billion protein sequence fragments predicted by Prodigal[19] in 2200 metagenomic and metatranscriptomic datasets[3,20,21] downloaded mainly from the Joint Genome Institute. We clustered these sequences with a minimum sequence identity of ≥50% and minimum coverage of the shorter sequence of 90% (Methods: Clustering criteria), producing 424 million clusters in 10 h on a 2 × 14-core server.

Our Metaclust database of 424 million representative sequences will improve the sensitivity of profile sequence searches by increasing the diversity of the underlying multiple sequence alignments. It will thereby raise the fraction of annotatable sequences in genomic and metagenomic datasets[6,21]. It could also increase the number protein families for which reliable structures can be predicted de novo, as shown by Ovchinnikov et al.[22], who used an unpublished dataset of 2 billion metagenomic sequences. Metaclust should also allow us to predict more accurately the effects of mutations on proteins[23].

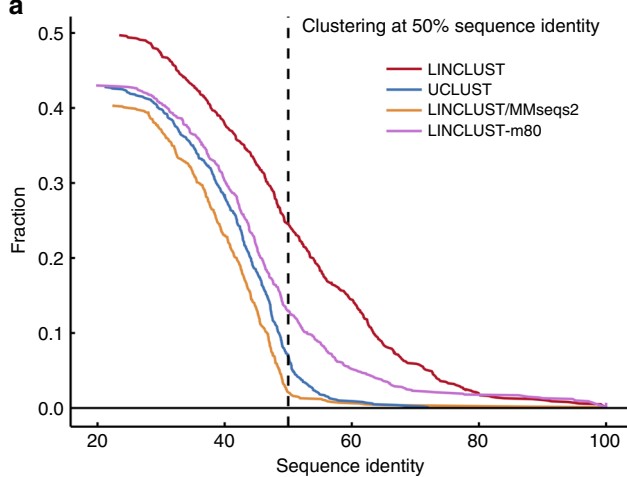

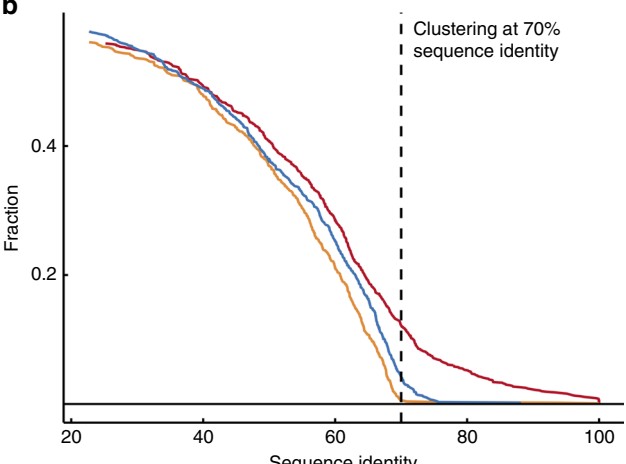

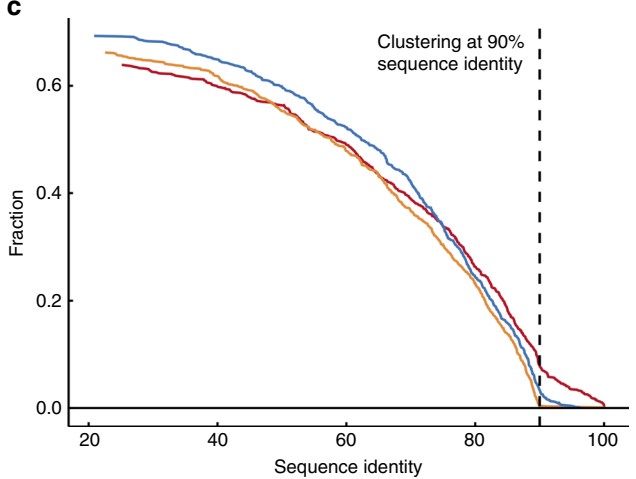

**Fig. 3** Cumulative distance distribution between representative sequences. We clustered the test set of 123 million sequences at three different sequence identity thresholds (**a**–**c** at 50%, 70%, and 90%, respectively). For each clustering, we randomly sampled 1000 representative cluster sequences, compared them to all representative sequences of the clustering, and plotted the fraction whose best match (excluding self-matches) with minimum sequence coverage of 90% had a sequence identity above the x-value. The y-value at the clustering threshold (dashed line) is the fraction of false negatives, pairs of sequences whose similarity was overlooked by the clustering method

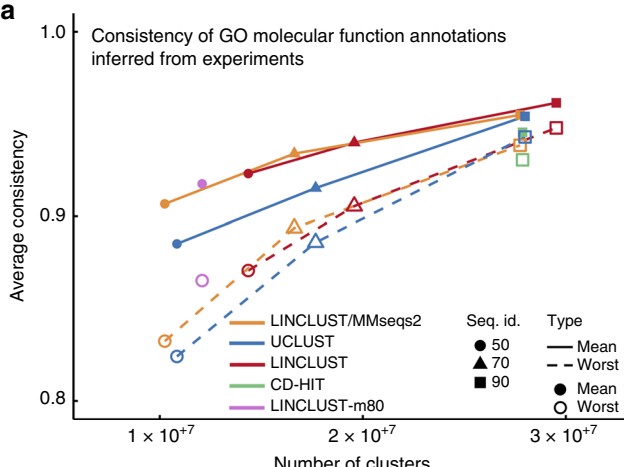

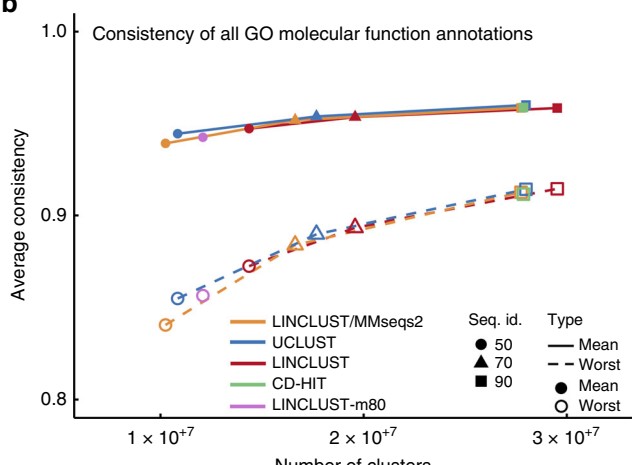

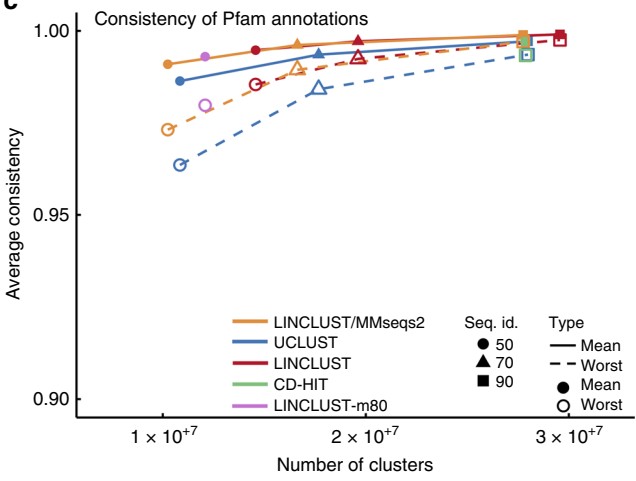

**Fig. 4** Cluster consistency of GO molecular functional and Pfam annotations. **a** Cluster annotation consistency of GO functional annotations inferred from experiments (EXP_F). "Mean" and "worst" refers to the mean and the minimum annotation similarity between each representative sequence and all other cluster members. Plotting symbols indicate the sequence identity threshold for clustering. CD-HIT was only run at 90% sequence identity due to run time constraints. Linclust-m80 was only run at 50% sequence identity. **b** Same as (**a**) but using manually and computationally assigned functional GO annotations. **c** Consistency of Pfam annotation from the representative sequences to the cluster members

## Discussion

Clustering a set of $N$ items is challenging when both $N$ and the number of clusters $K$ are large, due to the time complexity of existing clustering algorithms. Hierarchical agglomerative clustering approaches have a time complexity of $O(N^2\log N)$[24], others with a predefined number of clusters such as $K$-means or expectation maximization clustering have complexity $O(NK)$. When both $N$ and $K$ are in the tens of millions, traditional approaches are impracticably slow. Driven by the need to cluster huge datasets in the era of big data, most work has focused on reducing the proportionality constant.

One example is the widely used canopy clustering algorithm[25]. The items are first preclustered into overlapping sets ("canopies") based on a fast approximate similarity measure. Canopies could be biological sequences sharing the same $k$-mer or documents sharing a keyword. Some traditional clustering algorithm is run on all items, but with the restriction that slow, exact similarities are only computed between items belonging to the same canopy. Similar to the $k$-mer prefilter used in CD-HIT, UCLUST, kclust, and MMseqs[8,11,26,27], the preclustering reduces the number of comparisons by a large factor $F$ using the slow, exact measure, but the time complexity of the exact distance calculation $O(N^2/F)$ is still quadratic in $N$. Linear-time clustering algorithms, using for instance hashing techniques, have been proposed[28,29]. But like the preclustering step in canopy clustering or Linclust's prefilter to find $k$-mer matches, these algorithms are only approximate. If falsely clustered pairs are costly (e.g., for redundancy filtering), pairwise links need to be verified with the exact similarity measure, which still requires quadratic time complexity. In contrast, Linclust's linear time complexity of $O(mN)$ includes verification of all edges between items using the exact distance measure.

Linclust can be trivially generalized to cluster any items for which a set of $m$ keys per item can be defined such that (1) items belonging to a cluster are likely to share at least one of their keys and (2) items not belonging to a cluster are unlikely to share a key (see Methods, Optimal $k$-mer length). For clustering documents the keys could be all $m = \binom{n}{k}$ subsets of the $n$ keywords of size $k$, for example[28]. To achieve a high sensitivity, we could select as center of the group of items sharing a key the member with the largest sum of sizes of groups it belongs to. In this way, the center items are able to pull together into the same cluster many items from different groups.

We perform the clustering in step 5 of Fig. 1 with the greedy incremental clustering, because it always chooses the longest sequence as the cluster representative. It ensures that the representative sequences, being the longest sequence in each cluster, are likely to contain all protein domains of all cluster members. Our rule in step 2 to choose the longest protein sequence per $k$-mer group as its center is well-suited to achieve large clusters, because the longest sequences tend to be selected as centers of most of the $k$-mer groups they belong to, and these long sequences therefore have edges to most sequences they share $k$-mers with.

As far as we know, Linclust is the only algorithm that could run on datasets of billions of items resulting in billions of clusters, overcoming the time and memory bottlenecks of existing clustering algorithms. Linclust could therefore be useful for many other applications. We have recently extended Linclust to nucleotide sequences. We are also working on a version to cluster $D$-dimensional vectors, which could be used, for instance, for metagenomic binning to cluster contigs by their coverage profiles across $D$ metagenomic samples[30].

In summary, we hope the Linclust algorithm will prove helpful to exploit the tremendous value in publicly available metagenomic and metatranscriptomic datasets. Linclust should lead to considerable savings in computing resources in current

**Fig. 5** Linear-time clustering algorithm. Steps 1 and 2 find exact $k$-mer matches between the $N$ input sequences that are extended in step 3 and 4. (1) Linclust selects in each sequence the $m$ (default: 20) $k$-mers with the lowest hash function values, as this tends to select the same $k$-mers across homologous sequences. It uses a reduced alphabet of 13 letters for the $k$-mers and sets $k$ between 10 and 14 depending on the sequence set size and the sequence identity threshold. It generates a table in which each of the $mN$ lines consists of the $k$-mer, the sequence identifier, and the position of the $k$-mer in the sequence. (2) Linclust sorts the table by $k$-mer in quasi-linear time, which identifies groups of sequences sharing the same $k$-mer (large shaded boxes). For each $k$-mer group, it selects the longest sequence as center. It thereby tends to select the same sequences as center among groups sharing sequences. (3) It merges $k$-mer groups with the same center sequence together (**1**: red + cyan and **5**: orange + blue) and compares each group member to the center sequence in two steps: by global Hamming distance and by gapless local alignment extending the $k$-mer match. (4) Sequences above a score cut-off in step 3 are aligned to their center sequence using gapped local sequence alignment. Sequence pairs that satisfy the clustering criteria (e.g., on the $E$-value, sequence similarity, and sequence coverage) are linked by an edge. (5) The greedy incremental algorithm finds a clustering such that each input sequence has an edge to its cluster's representative sequence. Note that the number of sequence pairs compared in steps 3 and 4 is less than $mN$, resulting in a linear time complexity

applications. Most importantly, it will enable previously infeasible large-scale analyses.

## Methods
The Linclust algorithm consists of the following steps (Figs 1 and 5):

**Step 1: Generating the table of $k$-mers**. We transform the sequence set into a reduced alphabet of 13 letters to increase the number of $k$-mer matches and hence the $k$-mer sensitivity at the cost of a moderate reduction in selectivity (see subsection "Reduced amino acid alphabet"). The $k$-mer length is chosen as described in subsection "Optimal $k$-mer length" and is typically between 10 and 14.

For each sequence, we extract $m$ $k$-mers, as described in "Selection of $k$-mers". Increasing $m$ from its default value of 20 (option—kmer-per-seq) increases the sensitivity at the cost of a moderately decreasing speed (Supplementary Fig. S4). We store each extracted $k$-mer index (8 bytes), the sequence identifier (4 bytes), its length (2 bytes), and its position $j$ in the sequence (2 bytes) in a table with $mN$ lines. Therefore, Linclust has a memory footprint of $mN \times 16$ bytes.

**Step 2: Finding exact $k$-mer matches**. We sort this table by the $k$-mer index using the in-place sort from the OpenMP template library (http://freecode.com/projects/omptl). The sorting has a quasi-linear time complexity of $O(mN\log(mN))$ and typically takes less than 10% of the total runtime. The sorting groups together sequences into blocks of lines that contain the same $k$-mer. For each such $k$-mer group we select the longest sequence as its center sequence. We overwrite the position $j$ with the diagonal $i-j$ of the $k$-mer match with the center sequence, where $i$ is the position of the group's $k$-mer in the center sequence. We further overwrite the $k$-mer index by the center sequence identifier and resort the $mN$ lines of the table by the center sequence identifier. The $k$-mer match stage results file has one entry for each center sequence identifier containing the list of identifiers of

sequences that share a $k$-mer with the center sequence. If a sequence shares multiple $k$-mer matches with a center sequence, we keep only the entry with the lowest diagonal $i-j$.

**Step 3a: Hamming distance pre-clustering**. For each $k$-mer group we compute the Hamming distance (the number of mismatches) in the full amino acid alphabet between the center sequence and each sequence in the group along the stored diagonals $i-j$. This operation is fast as it needs no random memory or cache access and uses AVX2/SSE4.1 vector instructions. Members that already satisfy the specified sequence identity and coverage thresholds on the entire diagonal are removed from the results passed to step 3b and are added to the cluster of their center sequence after step 5.

**Step 3b: Ungapped alignment filtering**. For each $k$-mer group we compute the optimal ungapped, local alignment between the center sequence and each sequence in the group along the stored diagonals $i-j$, using one-dimensional dynamic programming with the Blosum62 matrix. We filter out matches between center and member sequences if the ungapped alignment score divided by the length of the diagonal is very low. We set a conservative threshold, such that the false negative rate is 1%, i.e., only 1% of the alignments below this threshold would satisfy the two criteria, sequence identity and coverage. For each combination on a grid {50, 55, 60, …,100}⊗{0, 10, 20,…,100}, we determined these thresholds empirically on 4 million local alignments sampled from an all-against-all comparison of the UniProt database[31].

**Step 4: Local gapped sequence alignment**. Sequences that pass the ungapped alignment filter are aligned to their center sequence using the AVX2/SSE4.1-vectorized alignment module with amino acid compositional bias correction from MMseqs2[10], which builds on code from the SSW library[32]. Sequences satisfying the specified sequence identity and coverage thresholds are linked by an edge. These

edges (neighbor relationships) are written in the format used by MMseqs2 for clustering results.

**Step 5: Greedy incremental clustering**. This algorithm was already implemented for MMseqs[11]. Briefly, the file with the validated directed edges from center sequences to member sequences is read in and all reverse edges are added. The list of input sequences is sorted by decreasing length. While the list is not yet empty, the top sequence is removed from the list, together with all sequences still in the list that share an edge with it. These sequences form a new cluster with the top sequence as its representative.

**Reduced amino acid alphabet**. We iteratively constructed reduced alphabets starting from the full amino acid alphabet. At each step, we merged the two letters $\{a, b\} \longrightarrow a' = (a \text{ or } b)$ that conserve the maximum mutual information, $\mathrm{MI} = \sum_{x,y=1}^{A} p(x,y)\log_2(p(x,y)/p(x)/p(y))$. Here $A$ is the new alphabet size, $p(x)$ is the probability of observing letter $x$ at any given position, and $p(x, y)$ is the probabilities of observing $x$ and $y$ aligned to each other. These probabilities are extracted from the Blosum62 matrix. When $a$ and $b$ are merged into $a'$, for example, $p(a') = p(a) + p(b)$ and $p(a', y) = p(a, y) + p(b, y)$. The default alphabet with $A = 13$, which performed well over all tested clustering sequence identities from 50% to 100%, merges (L, M), (I, V), (K, R), (E, Q), (A, S, T), (N, D), and (F, Y).

**Optimal $k$-mer length**. For optimal results and efficiency, the majority of the sequences in $k$-mer groups should be homologous to their center sequence. In other words, the $k$-mers have to be specific enough for the size of the database, with larger databases requiring larger $k$. To automatically set a good value of $k$, a very conservative condition is to limit to 1 the expectation value $E_{\mathrm{FP}}$ of the number of sequences per $k$-mer group that are not homologous to their center sequence. $E_{\mathrm{FP}}$ is equal to the number $mN$ of $k$-mers selected in the entire sequence set times the probability $p_{\mathrm{match}}$ for one of those $k$-mers to match the $k$-mer of the $k$-mer group by chance. If the $k$-mers were not preselected by their hash function values, this probability would be approximately $1/A_{\mathrm{eff}}^k$, where $1/A_{\mathrm{eff}} = \sum_{a=1}^{A} p_a^2$ is the probability for two letters from the reduced alphabet of size $A$ to match by chance (1/8.7 for $A = 13$) and $p_a$ is the frequency of letter $a$ in the database. Due to the preselection, only a fraction $\sim m/L$ of the entire set of $k$-mers is used, where $L$ is the average sequence length. Therefore, the probability of two selected $k$-mers to match by chance is $L/(mA_{\mathrm{eff}}^k)$. The condition for the $k$-mer specificity is $1 \geq E_{\mathrm{FP}} = mNL/\left(mA_{\mathrm{eff}}^k\right) = NL/A_{\mathrm{eff}}^k$, and hence we demand $k \geq \lfloor\log(NL)/\log(A_{\mathrm{eff}})\rfloor =: k_{\mathrm{spec}}$. In Linclust, we set $k = \max\{k_{\mathrm{spec}}, k_{\mathrm{seqid}}\}$, with $k_{\mathrm{seqid}} = 14$ for a sequence identity clustering threshold $\geq 90\%$ and $k_{\mathrm{seqid}} = 10$ otherwise to ensures slightly higher efficiency for high sequence identities, for which longer $k$-mers are sufficiently sensitive.

**Selection of $k$-mers**. To be able to cluster two sequences together we need to find a $k$-mer in the reduced alphabet that occurs in both. Because we extract only a small fraction of $k$-mers from each sequence, we need to avoid picking different $k$-mers in each sequence. Our first criterion for $k$-mer selection is therefore to extract $k$-mers such that the same $k$-mers tend to be extracted from homologous sequences. Second, we need to avoid positional clustering of selected $k$-mers in order to be sensitive to detect local homologies in every region of a sequence. Third, we would like to extract $k$-mers that tend to be conserved between homologous sequences. Note that we cannot simply store a subset of $A^k m/L$ $k$-mers to be selected due to its sheer size.

We can satisfy the first two criteria by computing hash values for all $k$-mers in a sequence and selecting the $m$ $k$-mers that obtain the lowest hash values. Since appropriate hash functions can produce values that are not correlated in any simple way with their argument, this method should randomly select $k$-mers from the sequences such that the same $k$-mers always tend to get selected in all sequences. We developed a simple 16-bit rolling hash function with good mixing properties, which we can compute very efficiently using the hash value of the previous $k$-mer (Supplementary Fig. 5).

In view of the third criterion, we experimented with combining the hash value with a $k$-mer conservation score $S_{\mathrm{cons}}(x_{1:k}) = \sum_{i=1}^{k} S(x_i, x_i)/k$. This score ranks $k$-mers $x_{1:k}$ by the conservation of their amino acids, according to the diagonal elements of the Blosum62 substitution matrix $S(\cdot, \cdot)$. We scaled the hash function with a rectified version of the conservation score: hash$-$value$(x_{1:k})/\max\{1, S_{\mathrm{cons}}(x_{1:k}) - S_{\mathrm{offset}}\}$. Despite its intuitive appeal, we did not succeed in obtaining significant improvements and reverted to the simple hash function.

**Clustering datasets that do not fit into main memory**. Linclust needs $m \times 16$ bytes of memory per sequence. If the computer's main memory is too small, Linclust automatically splits the $k$-mer array into $C$ equal-sized chunks small enough to fit each into main memory (Supplementary Fig. 1). For each chunk index $c \in \{0,...,C-1\}$ we run Linclust steps 1 and 2 (Fig. 5) normally but extract only $k$-mers whose numerical index modulo $C$ yields a rest $c$. This way each of the $C$ runs builds up a $k$-mer table with only about $mN/C$ lines instead of $mN$, and

hence each run needs $C$ times less memory. Each run writes out a file with all found $k$-mer groups, and afterwards all $C$ files are merged into a single file such that $k$-mer groups are sorted by ascending center IDs. Finally, Linclust steps 3 to 5 are performed as usual.

**Parallelization and supported platforms**. We used OpenMP to parallelize all stages except the fast step 5 and SIMD instructions to parallelize step 3 and step 4. Linclust supports Linux and Windows, Mac OS X and CPUs with AVX2 or SSE4.1 instructions.

**Clustering criteria**. Linclust/MMseqs2 and Linclust has three main criteria to link two sequences by an edge: (1) a maximum E-value threshold (option -e $[0, \infty[$) computed according to the gap-corrected Karlin-Altschul statistics using the ALP library;[33] (2) a minimum coverage (option -c $[0,1]$, which is defined by (a the number of aligned residue pairs divided by either the maximum of the length of query/ center and target/non-center sequences (default mode, --cov-mode 0), or by the length of the target/non-center sequence (--cov-mode 1), or by the length of the query/center (--cov-mode 2); (3) a minimum sequence identity (--min-seq-id $[0, 1]$) with option --alignment-mode 3 defined as the number of identical aligned residues divided by the number of aligned columns including internal gap columns, or, by default, defined by a highly correlated measure, the equivalent similarity score of the local alignment (including gap penalties) divided by the maximum of the lengths of the two locally aligned sequence segments. The score per residue equivalent to a certain sequence identity is obtained by a linear regression using thousands of local alignments as training set (Fig. S2 in[27]).

The sequence identity in UCLUST is defined as number of identical residues in the pairwise global alignment divided by the number of aligned columns including internal gaps. Due to the global alignment, no explicit coverage threshold is needed. CD-HIT defines sequence identity as the number of identical residues in the local alignment divided by the length of the shorter sequence. Therefore, sequence coverage of the shorter sequence must be at least as large as the sequence identity threshold.

**Tools and options for benchmark comparison**. Linclust and Linclust/MMseqs2 (commit 5e21868) used the commands mmseqs linclust --cov-mode 1 -c 0.9 --min-seq-id 0.9 and mmseqs cluster --cov-mode 1 -c 0.9 --min-seq-id 0.9 for 90%, respectively, and --min-seq-id 0.7 or --min-seq-id 0.5 for 70% and 50%. The minimum coverage of 90% of the shorter sequence was chosen to enforce global similarity, similar to UCLUST and CD-HIT. CD-HIT 4.6 was run with the parameters -T 16 -M 0 and -n 5 -c 0.9, -n 4 -c 0.7, and -n 3 -c 0.5 for 90%, 70%, and 50%, respectively. UCLUST (7.0.1090) was run with --id 0.9, 0.7, 0.5, for RAPsearch2 (2.23) we used -z 16, for DIAMOND (v0.8.36.98) option --id 0.5, and for MASH (v2.0) -s 20 -a -i -p 16. Runtimes were measured with the Linux time command.

**Functional consistency benchmark**. We evaluated the functional cluster consistency based on Gene Ontology (GO) annotations of the UniProt knowledge base. We carried out three tests: one based on (1) experimentally validated GO annotations, (2) general functional GO annotations (mostly inferred from homologous proteins) and (3) Pfam annotations. The UniProt 2016_03 release was clustered by each tool at 90%, 70% and 50% sequence identity level and then evaluated. For CD-HIT we computed only the clustering at 90% sequence identity because of run time constraints. For each cluster, we computed the 'worst' and 'mean' cluster consistency scores, as described earlier[16]. These cluster consistency scores are defined respectively as the minimum and the mean of all pairwise annotation similarity scores between the cluster's representative sequence and the other sequences in the cluster.

GO annotations often annotate the whole sequence. We used the Pfam annotations of the UniProt to check local consistence of clusters (Fig. 3c). We compared the Pfam domain annotation of the representative sequence against all cluster members. If the member had the exact same domain annotation as the representative sequence we counted it as correct (value = 1) and otherwise as false (value = 0).

**Clustering**. We downloaded ~1800 metagenomic and ~400 metatranscriptomic datasets with assembled contigs from the Joint Genome institute's IMG/M archive[3] and NCBI's Sequence Read Archive[20] (ftp://ftp.ncbi.nlm.nih.gov/sra/wgs_aux) using the script metadownload.sh from https://bitbucket.org/martin_steinegger/linclust-analysis. We predicted genes and protein sequences using Prodigal[19] resulting in 1,595,926,152 proteins.

We clustered the 1.59 million sequence fragments with Linclust using the following acceptance criteria: (1) the minimum sequence identity is 50%, using the score-per-column similarity measure described in Clustering criteria, (2) the shorter of the two sequences has at least 90% of its residues aligned, and (3) the maximum E-value is $10^{-3}$ (default) (Linclust options: --min-seq-id 0.5 --cov-mode 1 -c 0.9 --cluster-mode 2). The clustering step found 424 million cluster within 10 h on a server with two 14-core Intel Xeon E5-2680 v4 CPUs (2.4 GHz) and 762 GB RAM.

**Metaclust protein sequence sets**. The Metaclust database is available as FASTA formatted file at https://metaclust.mmseqs.org/.

**Code availability**. Linclust has been integrated into our free GPLv3-licensed MMseqs2 software suite[10]. The source code and binaries for Linclust can be download at https://github.com/soedinglab/mmseqs2.

**Data availability**. The Metaclust dataset generated during the current study is available at https://metaclust.mmseqs.org. The Linclust source code is available at https://mmseqs.org. All scripts and benchmark data including command-line parameters necessary to reproduce the benchmark and analysis results presented here are available at https://bitbucket.org/martin_steinegger/linclust-analysis.

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

## Acknowledgements

We are grateful to Cedric Notredame and Chaok Seok for hosting MS at the CRG in Barcelona and at Seoul National University for 12 and 30 months, respectively. Thanks to Milot Mirdita and Clovis Galiez for discussions and to all who contributed metagenomic datasets used to build Metaclust, in particular the US Department of Energy Joint Genome Institute http://www.jgi.doe.gov/ and their user community. This work was supported by the EU's Horizon 2020 Framework Programme (Virus-X, grant 685778).

## Author contributions

M.S. performed the research and programming, M.S. and J.S. jointly designed the research and wrote the manuscript.

## Additional information

**Competing interests:** The authors declare no competing interests.

