## [Peer Review File · Nature Communications]

Reviewers' comments:

From the editor: Please note that we have asked Reviewers 1 and 2 to comment on your responses to Reviewer 3's points. Reviewer 1 feels these points are fully addressed. Reviewer 2's comments are included in his/her Remarks to the Author below.

Reviewer #1 (Remarks to the Author):

This is a revised version of a manuscript I reviewed earlier for Nature Methods, in which the authors present Linclust, a linear time method to cluster sequences—a fundamental problem in bioinformatics.

The authors have addressed my comments satisfactorily (reviewer 1 in the original submission). In particular, the inclusion of MASH in the analysis is informative. I was aware that it had quadratic complexity due to the all-against-all configuration, but I imagined that the constant was much lower due to it being alignment-free. Turns out it is not even close when dealing with millions of sequences. I think this will make the method attractive.

The only remaining point I have is that, the authors should acknowledge in the paper that (in the authors's own words, from their point-by-point reply "neither Linclust, MMseqs2, nor UCLUST nor any other tool we know of is suitable to cluster down to 70% or 50% sequence identity, since they all use fast prefilters that can miss some similar pairs").

Such caveat would not take anything away from their impressive achievement, while at the same time avoid setting unrealistic expectations.

Reviewer #2 (Remarks to the Author):

I thank the authors for their clear answers and great work with Linclust and MMseqs2. The reformatted manuscript reads better, and some important points have been addressed.

Point 1. Sensitivity

I understand that sensitivity is the price to pay for clustering speed. What concerned me in the first place was that that price might be too high based on the data shown. Consider the following scenario: when clustering at 50% identity, one would theoretically expect that all sequences sharing 50% sequence identity are clustered into a single group. Any deviation from that, i.e. split clusters, could be considered an error. Let's now assume that MMseqs2 (or CD-HIT) are providing a perfect result. If, at the same identity threshold, Linclust produces twice as many clusters as MMseqs2 (as originally shown in Fig2B), it is inevitable to think that something went wrong. Yes, it ran much faster, but the obtained clusters might be so distant from the expected theoretical result that we can no longer call them 50%-identity clusters.

I completely agree with the authors that the obtained result would still be of great value (in many cases, the alternative would simply be having no results), but it is highly misleading to assume that the obtained clusters are at least 50% (identity) distant with each other, when they are not. The problem could simply be a semantic question, but it has deep implications in the post analysis of clusters: users will need to know that when MMseqs2/LinClust groups sequences at 50% identity, it actually means 50%+-10% or 70%+-20%, etc. Also, it would be crucial to know if the error rate increases linearly when relaxing identity. A simple post-clustering analysis using some of the small benchmarking datasets could perhaps provide an idea of what is the expected standard deviation or confidence value for the different identity thresholds (i.e. avg, max and min identity among resulting clusters).

The new clustering workflow combining LinClust and MMSeqs2 cascade method seems to rescue many missing links in LinClust clusters, therefore reducing the number of unexpected splits. This is great, however, knowing that results can be so different depending on the clustering strategy chosen, I would still consider necessary (and extremely useful) to know what is the standard deviation of each identity threshold when using the "mmseqs cluster" command. In my opinion, that would be much more informative than the functional consistency analysis.

Point 2. Incremental changes

LinClust novelty is now clearer in the paper. I still find weird that LinClust and the recently published MMseqs2 are considered two separate tools (especially now that the default clustering workflow is based in a combination of both algorithms), but they are anyways two very useful methods and they can be described in two separate papers.

Point 3. Nucleotide version

I can understand this might be too much work for a review. I really hope we can see those changes in future updates.

- Remarks on Reviewer 3's comments and the authors' responses

Apart from the novelty issues regarding MMseqs2 and LinClust overlap, I think all these comments are minor and mostly addressed.

Specific comments:

One of the frequent problems encountered when clustering sequence databases, is the phenomenon of domain chaining. The authors have used GO-terms to try and assess the functional congruence of the clusters. However, as GO-terms are typically applied to the function of the whole sequence, they can often mask the subtle effect of domain chaining. The parameters used for clustering are likely to negate much of this, but it would be interesting to consider the consistency of the clusters in terms of, say, Pfam domains.

A Pfam analysis has been included in the manuscript, showing that MMSeqs achieves better consistency than other clustering methods. As mentioned in the paper, this is most likely due to the coverage threshold imposed by LinClust (please specify in the paper). Nevertheless, I don't think this sheds much light to the problem of clustering multidomain proteins. The three different clustering strategies and coverage calculation methods implemented in MMseqs2 could indeed be used to investigate this further but, again, this looks a MMSeqs2 feature rather than part of the LinClust clustering method.

It is also a little strange to have an article describing a sub-part of the suite of programs. I believe I understand the reason why from a software engineering point of view, but it feels - to some extent -

like an iterative improvement of the MMSeqs software that was published only last year. This has been partially addressed by focusing on the novelty of the clustering method, although MMseqs2 features and Linclust are, in practice, working together.

Nevertheless, I have reviewed many informatics papers where the performance never quite meets expectation. However, in this case I would like to congratulate the authors on the speed of the software. The software installs smoothly, although I would point out that the build commands on GitHub mentions generating a build directory, changing into that directory and making from there, but you need to be in the MMseq2 directory. I think having a dedicated section for the use of Linclust in the README is essential.

The software builds correctly under Linux systems and precompiled binaries are provided. The documentation is now clear.

The current examples demonstrate clustering using mmseqs. I needed to guess quite a few of the steps. While it appears to have clustered my fairly large sequence database (~110 million -> 79 million clusters) – the clusters look strange when trying to decompose them back out into the sets of sequence that have been grouped into the cluster.

I guess this comment was done assuming that the result file was a FASTA file, but it is not. The referred format is MMseqs' internal format. It would indeed be useful that MMseqs provides an option to export all or selected clusters as a collection of canonical FASTA files.

Finally, it would be interesting to comment on how frequently the Metaclust data will be updated in the future, if at all?

This is still not mentioned, but probably not necessary as the focus of the paper has shifted towards method description.

23 March 2018

Transfer of manuscript NCOMMS-18-00387-T from Nature Methods

Dear Editor,

we have addressed all remaining reviewer comments below and highlighted all changes in the manuscript in blue. We added a new Figure 1 that explains on a simpler level how Linclust achieves linear run time and we moved the more detailed old Figure 1 to the methods section as Figure 5. Motivated by a reviewer comment about the number of missed similarities in Linclust, we also added an analysis of false negatives in Figure 3a,b, and c. We also added a paragraph “Optimal k -mer length” to the methods section that explains how Linclust automatically sets k as a function of the database size.

Reviewer #1 (Remarks to the Author):

This is a revised version of a manuscript I reviewed earlier for Nature Methods, in which the authors present Linclust, a linear time method to cluster sequences—a fundamental problem in bioinformatics.

The authors have addressed my comments satisfactorily (reviewer 1 in the original submission). In particular, the inclusion of MASH in the analysis is informative. I was aware that it had quadratic complexity due to the all-against-all configuration, but I imagined that the constant was much lower due to it being alignment-free. Turns out it is not even close when dealing with millions of sequences. I think this will make the method attractive.

The only remaining point I have is that, the authors should acknowledge in the paper that (in the authors's own words, from their point-by-point reply "neither Linclust, MMseqs2, nor UCLUST nor any other tool we know of is suitable to cluster down to 70% or 50% sequence identity, since they all use fast prefilters that can miss some similar pairs".

Such caveat would not take anything away from their impressive achievement, while at the same time avoid setting unrealistic expectations.

Thank you for your positive assessment. We quantified the number of sequences that would satisfy the clustering criteria but are missed by clustering tools (Figure 3) We avert in the main text to the fact that all methods overlook a non-negligible fraction of true links at 50% sequence identity.

Reviewer #2 (Remarks to the Author):

I thank the authors for their clear answers and great work with Linclust and MMseqs2. The reformatted manuscript reads better, and some important points have been addressed.

Point 1. Sensitivity

I understand that sensitivity is the price to pay for clustering speed. What concerned me in the first place was that that price might be too high based on the data shown. Consider the following scenario: when clustering at 50% identity, one would theoretically expect that all sequences sharing 50% sequence identity are clustered into a single group. Any deviation from that, i.e. split clusters, could be considered an error. Let's now assume that MMseqs2 (or CD-HIT) are providing a perfect result. If, at the same identity threshold, Linclust produces twice as many clusters as MMseqs2 (as originally shown in Fig2B), it is inevitable to think that something went wrong. Yes, it ran much faster, but the obtained clusters might be so distant from the expected theoretical result that we can no longer call them 50%-identity clusters.

I completely agree with the authors that the obtained result would still be of great value (in many cases, the alternative would simple be having no results), but it is highly misleading to assume that the obtained clusters are at least 50% (identity) distant with each other, when they are not. The problem could simply be a semantic question, but it has deep implications in the post analysis of clusters: users will need to know that when MMseqs2/Linclust groups sequences at 50% identity, it actually means 50%+-10% or 70%+-20%, etc. Also, it would be crucial to know if the error rate increases linearly when relaxing identity. A simple post-clustering analysis using some of the small benchmarking datasets could perhaps provide an idea of what is the expected standard deviation or confidence value for the different identity thresholds (i.e. avg, max and min identity among resulting clusters).

The new clustering workflow combining Linclust and MMSeqs2 cascade method seems to rescue many missing links in Linclust clusters, therefore reducing the number of unexpected splits. This is great, however, knowing that results can be so different depending on the clustering strategy chosen, I would still consider necessary (and extremely useful) to know what is the standard deviation of each identity threshold when using the "mmseqs cluster" command. In my opinion, that would be much more informative than the functional consistency analysis.

Thank you for this detailed explanation of the issue you had, which we did not fully understand from the previous review. We agree that the issue of false negatives, of which we were not much aware due to the applications we had in mind, might actually be shared by many users. We have included the analysis of the clusters you suggested: We sample 1000 clusters from each of the obtained clusterings (at 90%, 70% and 50% sequence identity) and searched (using BLAST) with the representative sequences through all representative sequences of the clustering. We then recorded the cumulative distribution of best matches over sequence identity (x axis). When x is equal to the sequence identity used for the clustering, the value on the y axis gives the fraction of missed links. The results show that Linclust/MMseqs2 even at 50% misses only 2% of the links, whereas Linclust -m 80 and Linclust overlook 13% and 25%.

Point 2. Incremental changes

Linclust novelty is now clearer in the paper. I still find weird that Linclust and the recently published MMseqs2 are considered two separate tools (especially now that the default clustering workflow is based in a combination of both algorithms), but they are anyways two very useful methods and they can be described in two separate papers.

Our point is that they are not so much separate tools as separate algorithms. MMseqs2 is a search algorithm that can be used to compute similarity graphs, on which a simple greedy clustering algorithm can be run. Linclust is the first algorithm to cluster in linear time (including checking the links), which we expect to become an enabling algorithm for metagenomics. (It already is in our own lab.) That it makes sense to combine Linclust with a previously published method into a new tool does not diminish the novelty and power of the Linclust algorithm. It would have been bad for users if we had subordinated the usability of the tools to marketing consideration by creating a separate repository for Linclust to avoid the critique of too much overlap with MMseqs2.

Point 3. Nucleotide version

I can understand this might be too much work for a review. I really hope we can see those changes in future updates.

We implemented a nucleotide clustering version of Linclust and made it available as part of the open-source mmseqs2 distribution. It required developing a banded nucleotide alignment module and several other extensions. We reproduced the CD-HIT clustering from the "Structure and function of the global ocean microbiome" study by Sunagawa et al. Our clustering of the 111 million predicted genes resulted in a similar number of clusters as the ~39 million obtained by Sunagawa et al. The clustering took less than an hour on a 2x8 core machine. These are just preliminary results which we did not include into the manuscript.

- Remarks on Reviewer 3's comments and the authors' responses

Apart from the novelty issues regarding MMseqs2 and Linclust overlap, I think all these comments are minor and mostly addressed.

Specific comments:

> One of the frequent problems encountered when clustering sequence databases, is the phenomenon of domain chaining. The authors have used GO-terms to try and assess the functional congruence of the clusters. However, as GO-terms are typically applied to the function of the whole sequence, they can often mask the subtle effect of domain chaining. The parameters used for clustering are likely to negate much of this, but it would be interesting to consider the consistency of the clusters in terms of, say, Pfam domains.

A Pfam analysis has been included in the manuscript, showing that MMSeqs achieves better consistency than other clustering methods. As mentioned in the paper, this is most likely due to the coverage threshold imposed by Linclust (please specify in the paper). Nevertheless, I don't think this sheds much light to the problem of clustering multidomain proteins. The three different clustering strategies and coverage calculation methods implemented in MMseqs2 could indeed be used to investigate this further but, again, this looks a MMSeqs2 feature rather than part of the Linclust clustering method.

> It is also a little strange to have an article describing a sub-part of the suite of programs. I believe I understand the reason why from a software engineering point of view, but it feels - to some extent - like an iterative improvement of the MMSeqs software that was published only last year.

This has been partially addressed by focusing on the novelty of the clustering method, although MMseqs2 features and Linclust are, in practice, working together.

> Nevertheless, I have reviewed many informatics papers where the performance never quite meets expectation. However, in this case I would like to congratulate the authors on the speed of the software. The software installs smoothly, although I would point out that the build commands on GitHub mentions generating a build directory, changing into that directory and making from there, but you need to be in the MMseq2 directory. I think having a dedicated section for the use of Linclust in the README is essential.

The software builds correctly under Linux systems and precompiled binaries are provided. The documentation is now clear.

> The current examples demonstrate clustering using mmseqs. I needed to guess quite a few of the steps. While it appears to have clustered my fairly large sequence database (~110 million -> 79 million clusters) – the clusters look strange when trying to decompose them back out into the sets of sequence that have been grouped into the cluster.

I guess this comment was done assuming that the result file was a FASTA file, but it is not. The referred format is MMseqs' internal format. It would indeed be useful that MMseqs provides an option to export all or selected clusters as a collection of canonical FASTA files.

In the mmseqs2 documentation on <https://github.com/soedinglab/MMseqs2/wiki#clustering-format> we explain how to generate a fasta like format containing all sequences for a clustering.

```
$ mmseqs createseqfiledb DB DB_clu DB_clu_seq  
$ mmseqs result2flat DB DB DB_clu_seq DB_clu_seq.fasta
```

We also developed a workflow "mmseqs easy-cluster" which, by default, writes files containing all sequences for each cluster (all_seqs.fasta), the representative sequences (rep_seq.fasta) and the identifiers for all cluster members (cluster.tsv)

We would like to thank the reviewers for their time to test our software and provide detailed feedback to the manuscript, which helped to improve it a lot.

Sincerely,
Johannes and Martin

REVIEWERS' COMMENTS:

Although Reviewer 2 doesn't have Remarks to the Author, in Remarks to the Editor, he/she feels all his/her comments have been addressed